# Receptor Binding for the Entry Mechanisms of SARS-CoV-2: Insights from the Original Strain and Emerging Variants

**DOI:** 10.3390/v17050691

**Published:** 2025-05-10

**Authors:** Mohamed Mahdi, Irene Wanjiru Kiarie, János András Mótyán, Gyula Hoffka, Aya Shamal Al-Muffti, Attila Tóth, József Tőzsér

**Affiliations:** 1Department of Biochemistry and Molecular Biology, Faculty of Medicine, University of Debrecen, 4032 Debrecen, Hungary; kiarie.irene@med.unideb.hu (I.W.K.); motyan.janos@med.unideb.hu (J.A.M.); hoffka.gyula@med.unideb.hu (G.H.); aya.muffti@med.unideb.hu (A.S.A.-M.); 2Department of Infectology, Faculty of Medicine, University of Debrecen, 4031 Debrecen, Hungary; 3Doctoral School of Molecular Cellular and Immune Biology, University of Debrecen, 4032 Debrecen, Hungary; 4Department of Chemistry, Lund University, Box 124, 221 00 Lund, Sweden; 5Division of Clinical Physiology, Department of Cardiology, Faculty of Medicine, University of Debrecen, 4032 Debrecen, Hungary; atitoth@med.unideb.hu

**Keywords:** coronaviruses, SARS-CoV-2, viral entry, receptor utilization, COVID-19

## Abstract

Since its emergence in late 2019, severe acute respiratory syndrome coronavirus 2 (SARS-CoV-2) has continuously evolved, giving rise to multiple variants that have significantly altered the trajectory of the COVID-19 pandemic. These variants have resulted in multiple waves of the pandemic, exhibiting characteristic mutations in the spike (S) protein that may have affected receptor interaction, tissue tropism, and cell entry mechanisms. While the virus was shown to primarily utilize the angiotensin-converting enzyme 2 (ACE2) receptor and host proteases such as transmembrane serine protease 2 (TMPRSS2) for entry into host cells, alterations in the S protein have resulted in changes to receptor binding affinity and use of alternative receptors, potentially expanding the virus’s ability to infect different cell types or tissues, contributing to shifts in clinical presentation. These changes have been linked to variations in disease severity, the emergence of new clinical manifestations, and altered transmission dynamics. In this paper, we overview the evolving receptor utilization strategies of SARS-CoV-2, focusing on how mutations in the S protein may have influenced viral entry mechanisms and clinical outcomes across the ongoing pandemic waves.

## 1. Introduction

Coronaviruses (CoVs) represent a family of RNA viruses that affect humans and a broad range of animal species, inducing a spectrum of diseases ranging from mild respiratory infections to severe life-threatening conditions [1]. Over the years, CoVs have undergone significant evolutionary adaptations, enhancing their infectivity and transmissibility. They have been responsible for multiple significant outbreaks in the last two decades, notably, severe acute respiratory syndrome (SARS) in 2002–2003, Middle East respiratory syndrome (MERS) in 2012, porcine epidemic diarrhea coronavirus (PEDV) outbreak in 2013, and recently, the coronavirus disease of 2019 (COVID-19) pandemic [2]. Given their high reproduction rates and accelerated evolutionary trends, these viruses are capable of crossing species barriers and pose a significant public health burden due to their frequent zoonotic transmission.

Among the four identified coronavirus genera, human coronaviruses (HCoVs) belong to the Alpha and Beta coronavirus genera [3]. Seven coronavirus strains are known to cause diseases in humans, with four strains—229E, NL63, OC43, and HKU1—causing seasonal mild infections [4,5,6,7], while SARS-CoV, MERS-CoV, and SARS-CoV-2 may lead to the development of severe and potentially lethal infections, often causing recurring pandemics and constituting great social and economic burdens [8,9,10]. Bats and other intermediate hosts are considered to be the source of the three virulent viruses due to congruence in the genetic constitution [11]. For instance, SARS-CoV depicts a 99% sequence identity with a virus infecting civet cats [12], while MERS-CoV was transmitted to humans by dromedary camels [13]. SARS-CoV-2 shows 98% sequence identity to the bat coronavirus strain RaTG13 and to human SARS-CoV with 79% genetic identity [14].

Compared to DNA viruses, RNA viruses, like coronaviruses, have a high rate of mutation, with vital consequences for fast adaptation and cross-species transmissions [15,16,17]. Frequent human contact with wildlife and domesticated animals has played a significant role in their evolution and in the selection of mutations that favor receptor adaptability, moreover, the continuous exposure to animal reservoirs allows these pathogens to test various receptor interactions until they evolve mutations that enable the cross-species jump [18,19].

Key aspects of coronavirus permissivity, pathogenicity, and host specificity are their mechanism of entry, which is mediated by interactions between the viral spike (S) protein and specific receptors on host cell surfaces. This entry process is critical for viral infection, initiating the fusion between viral and host membranes, allowing the viral genome to enter host cells [20]. The entry receptor-binding mechanism involves complex interactions of the receptor-binding domains (RBDs) of the S proteins, which undergo conformational shifts to achieve optimal binding affinity with host target receptors [20]. The S protein is a trimeric transmembrane protein comprising two subunits, S1 and S2. S1 contains the RBD that recognizes and binds the target receptor on host cells, while the S2 subunit enables viral fusion with the host cell membrane upon priming by host proteases; such as transmembrane protease serine 2 (TMPRSS2) [21,22]. The role of cellular proteases in priming viral glycoproteins for entry is not unique to SARS-CoVs. Like the SARS-CoV-2 S protein, the influenza virus utilizes its hemagglutinin (HA) protein to recognize and bind host cell receptors, typically sialic acid residues. Following receptor binding, both viruses require host proteases for priming; SARS-CoV-2 depends on TMPRSS2 or cathepsins at the entry process, while influenza HA undergoes cleavage by host proteases such as trypsin or furin before assembly. This cleavage triggers conformational changes that facilitate membrane fusion, allowing viral RNA to enter the cytoplasm for replication [23]. Thus, while the specific receptors and fusion mechanisms differ, both viruses exploit host proteases and glycoprotein-mediated entry to infect cells.

RBD of the S protein is known to undergo mutations that can significantly influence host specificity and infectivity. Research shows that small mutations within the RBD can allow SARS-like coronaviruses to switch hosts, and to bind to human angiotensin-converting enzyme 2 (ACE2), despite their animal origin [17]. For example, several variants of SARS-CoV-2 have emerged with mutations in the RBD that increase the binding efficiency to ACE2, enhancing their ability to spread between humans and potentially cross to new hosts [24].

Coronaviruses exhibit a notable adaptability in receptor usage, exploiting both proteins and carbohydrate molecules on the cell surface. Commonly used and well-characterized receptors include ACE2, dipeptidyl peptidase 4 (DPP4), and aminopeptidase N (APN), utilized, respectively, by SARS-CoV-1 and -2, MERS-CoV, and HCoV-229E. Some coronaviruses use alternative receptors, such as sugars and cell adhesion molecules [25]. Coronaviruses HKU1 and OC43 employ 9-O-acetylated sialic acids for attachment [26], while heparin sulfate is used by NL63 for attachment in addition to ACE2 for cell entry [27]. HKU1 uses TMPRSS2 as a protein receptor, while there is no known protein receptor for OC43 [28]. A summary of human pathogenic coronaviruses, including their associated diseases, so far identified primary receptors, and major outbreak years, is presented in Table 1.

Compared to non-enveloped viruses, enveloped viruses exhibit a greater potential for transmission across different host species. Zoonotic viruses, which belong to diverse viral families, include SARS-CoV-2 (Coronaviridae), influenza (Orthomyxoviridae), and HIV (Retroviridae) [33]. This increased potential for evolving towards the use of novel receptors is attributed to the relatively low rigidity of the envelope proteins. The ability to utilize alternative receptors expands the range of host cells and species, thereby increasing the risk of cross-species transmission and zoonosis. A correlation between the number of alternative receptors and the host range has been established, with various proteins serving as receptors for entry across diverse viral families [34].

## 2. Entry Receptors for Coronaviruses

Human and animal coronaviruses employ several key receptors, each facilitating distinct viral entry pathways and influencing the virus-host interactions. Here, we will explore the main receptors involved in the attachment and entry of coronaviruses, focusing on SARS-CoV-2.

### 2.1. ACE2 (Angiotensin-Converting Enzyme 2)

ACE2 is a multifunctional protein that is, in its full length, a type I transmembrane protein comprising two domains: an extracellular amino terminal containing a HEXXH zinc-binding metalloprotease, and a C-terminal collectrin-like domain (CLD) [35,36]. CLD is followed by a single transmembrane helix that anchors the protein to the cell membrane. Beyond the transmembrane region, the very end of the C-terminal domain consists of an intracellular segment, the length of which is approximately 40 amino acids long [37]. The membrane-bound enzyme functions primarily in the renin-angiotensin system (RAS), counter-regulating ACE protein, and thereby, regulating blood pressure and inflammation [38,39]. A circulating (soluble) form of ACE2 (sACE2) is formed as a result of cleavage from the full-length protein by the action of a disintegrin and metalloprotease 17 (ADAM17), which is subsequently released into the extracellular space [40]. Retaining its enzymatic activity, it is thought that sACE2 extends the protective actions of ACE2 to sites distant from its original location on the cell membrane. Additionally, it may serve as a decoy by binding to circulating viruses, thereby reducing viral entry into host cells, mitigating infection severity [41]. ACE2 is primarily expressed at both the mRNA and protein levels in vascular endothelium, enterocytes, renal tubular cells, gall bladder epithelial cells, cardiomyocytes, and among cells of the reproductive system. Interestingly, its basal expression is relatively low in the respiratory system [42,43,44]. While some studies reported an increased expression in the respiratory system of COVID-19 patients [45], recent studies were not able to verify the induction of ACE2 by SARS-CoV-2 [46].

ACE2 is the primary entry receptor for SARS-CoV and SARS-CoV-2, as well as other closely related bat coronaviruses [47,48]. Evidence indicates that ACE2 might play a dual role during infection, functioning not only as an attachment factor, but also as a facilitator for the conformational activation of the SARS-CoV-2 S protein, thereby enabling its fusion with the cell membrane. This is similar to what is observed during HIV infection, wherein virus attachment to the target receptors and interaction with the co-receptors triggers conformational activation of the envelope protein, mediating fusion [49]. Indeed, SARS-CoV-2 pseudovirions and virus-like particles were found to undergo membrane fusion in the absence of ACE2, provided they are activated by a suitable protease, proving that ACE2 is not necessary for the membrane fusion of SARS-CoV-2. Nevertheless, the addition of soluble ACE2 was found to enhance the speed of the fusion process [50].

The S protein of SARS-CoV-2 undergoes critical proteolytic cleavage at two distinct sites, essential for viral entry. First, at the S1/S2 site, the proprotein convertase furin cleaves the S protein before assembly and budding, separating it into the S1 (receptor-binding) and S2 (fusion) subunits. This cleavage primes the protein for subsequent fusion upon attachment to host receptors. The second cleavage occurs at the S2′ site, near the fusion peptide, and is mediated by the target cell’s surface protease TMPRSS2. This cleavage exposes the fusion peptide, enabling the S2 subunit to refold into its post-fusion state, facilitating membrane fusion. To summarize, in the classical entry model, the S protein undergoes structural rearrangements, exposing the RBD to interact with the ACE2 receptor. After attachment, the protein is further primed by furin at the S1/S2 site during maturation, enabling TMPRSS2-mediated cleavage at the S2′ site. This step activates the fusion peptide, allowing the spike’s S2 subunit to mediate fusion of the viral and host cell membranes, leading to the internalization of the viral genome [51,52,53,54,55,56].

RBD of the S protein in SARS-CoV-2 has evolved to have a high binding affinity to ACE2. Structural studies indicate that even slight variations in the amino acid sequence of ACE2 among species can alter the binding affinity, affecting the virus’s ability to infect different hosts [57]. RBD of coronaviruses is a 220-residue-long segment located in the S1 subunit of the S protein, comprising of residues 331–528 and containing two subdomains—a five-stranded anti-parallel beta sheet core, and an extended loop formed by a two-stranded Beta sheet (residues 424–494), termed receptor binding motif (RBM) that mediates contact with ACE2 at 14 positions—that are considered hotspots and key for binding [11,58,59]. In the case of SARS-CoV-2, among these 14 positions, 8 are fully conserved, while the remaining 6 are substituted with (semi)conservative variations, at least in the earlier variants of the virus [59] (Figure 1).

Subsequent variants of SARS-CoV-2 were shown to have variable affinity to ACE2 compared to the prototypical Wuhan type (wild-type) [61]. N501Y mutation in both Beta and Gamma variants was demonstrated to convey a higher affinity to ACE2 relative to the wild-type, and while lacking the N501Y mutation, the Delta variant possessed the L452R and T478K mutations that increased the RBD’s affinity to bind the receptor through alteration of hydrogen bonds and neutralization of hydrophobic patches in the binding pocket [54,61,62,63]. Other notable mutations in the RBD, such as K417N (Beta, Gamma), R346K (Mu), F490S (Lambda), and E484K (Beta, Gamma, Zeta, Eta, Theta, Iota, Mu), have been associated with enhanced immune evasion by reducing susceptibility to neutralizing antibodies. Additionally, K417N and E484K have been reported to modestly alter binding affinity to the ACE2 receptor, potentially influencing viral entry and infectivity [64,65]. A summary of key RBD mutations in SARS-CoV-2 variants is presented in Table 2.

In regard to the Omicron variant, mutations in residues N478, Q493, and Q498, in addition to the N501 mutation, are believed to be the reason behind the increased binding affinity of this variant to ACE2 [67].

The extensive mutations observed within the RBD of newer SARS-CoV-2 variants suggest the potential to modulate the binding affinity of the S protein to the ACE2 receptor in either direction, enhancing or diminishing interaction strength. Structural modeling and computational predictions indicate that specific amino acid substitutions may alter key molecular interactions, such as hydrogen bonds, electrostatic forces, and hydrophobic contacts, which collectively govern receptor engagement. In our modeling prediction, the incorporation of known mutations from the latest Omicron subvariants and other recently emerged lineages yielded intriguing results.

We have run AlphaFold3 [68] to predict the structure of the ACE2 receptor dimer in complex with the RBD region (three replicas of the 331–529 residue sequence) of the SARS-CoV-2 S protein variants, including the wild-type, JN.1, and XEC variants. As the AlphaFold software was previously applied to carry out the prediction of conformational distributions [69], we opted to apply it to estimate whether it reveals a difference in the tendency of correct binding of RBD to the ACE2 receptor. Our results reveal a stronger tendency for correct binding of the wild-type RBD region to ACE2 (60% of predicted complexes, Figure 2a), with a greatly reduced correct binding in case of the JN.1 variant at 32%, and an even lower tendency in case of the XEC variant at 20% (Figure 2b), with the majority of structures showing a misplaced RBD subunit, near the transmembrane domain of ACE2 (Figure 2c). Furthermore, to solidify our findings, these structures were then used as input for docking simulations with the GRAMM webserver [70]. The webserver was configured to generate 10 docked conformations for each case. Among these, only the wild-type variant produced a conformation closely resembling the correctly bound complex (Figure 2d), whereas the JN.1 and XEC variants resulted in inaccurately bound conformers. These findings underscore the complexity of RBD-ACE2 binding dynamics and highlight the necessity for further structural and functional validation to accurately assess the implications of these mutations on viral transmissibility.

### 2.2. Transmembrane Protease, Serine 2

The transmembrane protease, serine 2 (TMPRSS2), is a type 2 transmembrane member of the serine protease family. Structurally, it is composed of a type II hydrophobic transmembrane domain, a cysteine-rich extracellular domain, and a protease domain with the His, Asp, and Ser catalytic triad. Physiological functions of this protease include degradation and remodeling of the extracellular matrix, hemostatic regulation, in addition to activation of membrane proteins [71]. As mentioned previously, cleavage at the S2′ site is known to be a prerequisite for the release of the fusion peptide (FP) and subsequent fusion and entry of prototype SARS-CoV-2 [72]. TMPRSS2 is abundantly expressed in the respiratory epithelia, gastrointestinal system, liver, kidney, prostate, and epididymis, among other male tissues. Interestingly, it is almost absent from female reproductive organs [73]. Actions of this transmembrane protease have been implicated in the facilitation of infections of not just SARS-CoV-2, but also SARS-CoV, MERS, and other coronaviruses such as 229E, OC43, HKU1, influenza, as well as various other viruses [74,75,76,77]. Cleavage at the S2′ site by TMPRSS2 (positions 815–816) on the plasma membrane follows prior processing of the S1/S2 site by cellular furin, and places the FP at the newly formed N-terminus of the S2 subunit, initiating a significant and irreversible conformational shift that facilitates membrane fusion [78,79,80]. Studies and current literature data have shown that cleavage by both furin and TMPRSS2 is required for efficient prototypical SARS-CoV-2 infection of certain tissues, such as the lung epithelial cells [80].

Subsequent variants such as the Alpha and Delta harbored a mutation at the P681 (P681H and P681D, respectively) residue in the S1/S2 furin cleavage site, which was shown to increase proteolysis and enhance membrane fusion; however, whether or not that positively enhanced viral infectivity remains a matter of debate [81]. Interestingly, mutation of the residue R resulted in enhanced viral replication in VeroE6 cells overexpressing TMPRSS2 [82].

The Omicron variants are known to have over 30 mutations in the N-terminal and RBD, in residues N764, D796, N856 (observed only in the BA.1 variant), Q954, N969, and L981 that are located around the S2′ cleavage site heptad repeat 1 region [83]. Omicron variants retain the P681H mutation but exhibit a shift toward TMPRSS2-independent entry pathways [84], while the Gamma variant harbors H655Y, potentially influencing S processing and viral adaptability and fitness [85].

D614G, which is observed in Alpha, Beta, Gamma, Delta, and all current Omicron variants, enhances S stability and indirectly enhances the proteolytic processing by furin at the S1/S2 site, further facilitating viral adaptation and natural selection [86,87]. In single-cycle infection assays [88], the D614G mutation was shown to increase the stability of the spike protein’s S1/S2 interface. This enhanced stability, in turn, created a permissive environment for the acquisition of additional mutations that promoted more efficient cleavage at the S1/S2 site, ultimately driving the evolution of SARS-CoV-2 variants with highly fusogenic S proteins. The authors were able to corroborate previous findings indicating that increased S protein stability facilitates viral entry through the cathepsin L-dependent pathway [89].

A mutation in the N679 residue located next to the furin cleavage site, which has emerged in the Omicron and its subsequent subvariants (N679K), is also thought to enhance cleavage efficiency [90].

### 2.3. TMPSS2-Independent Pathways

We now know that entry of SARS-CoV-2 is not limited to the Furin-TMPRSS2 pathway, akin to other viruses and coronaviruses [91]. During this entry mechanism, termed TMPRSS2-independent, or late entry pathway, lysosomal cathepsin L plays a pivotal role. Cathepsin L is a papain-like cysteine protease that functions primarily in the turnover of intracellular and secreted proteins, degradation of pathogenic proteins, and antigen processing. Besides its dominant presence in lysosomes, cathepsin L is also secreted to process extracellular proteins [92,93]. In this pathway, cathepsin L was shown to enhance infectivity of SARS-CoV-2 by cleaving the S protein in at least two distinct sites at positions 255–264 and 632–641, in a region that possesses conservative sequences among different SARS-CoV-2 variants [94,95,96]. On one hand, cleavage was found to enhance the flexibility of the RBD, making it more accessible for binding to ACE2, and on the other hand, cleavage at the S1/S2 site near the furin cleavage site in the CTD separates S1 and S2 subunits, exposing the fusion peptide in the S2 subunit, and resulting in fusion between the viral envelope and the host cell membrane [94].

Utilization of either of the two pathways was found to be dependent on the variant as well as the expression level of TMPRSS2 in the target tissue. Earlier variants of SARS-CoV-2 were shown to predominantly rely on furin and the TMPRSS2 pathway for entry in airway epithelia and intestinal tissues [97,98], wherein expression of TMPSSS2 is abundant [99]. In regards to the Omicron variants, while many studies showed an inefficient cleavage of the Omicron S protein by the furin and TMPRSS2, indicating the primarily utilization of TMPRSS2-independent pathway, characterized by attenuated replication and pathogenicity [100,101], others were not able to detect a difference in cleavage of the S protein between the Omicron and earlier variants [94,102]. In previous studies, TMPRSS2 knockout in a murine model in mice, and using serine protease inhibitors such as nafamostat, significantly decreased the infectivity of Omicron variants [102].

The somewhat conflicting findings regarding Omicron’s use of TMPRSS2-dependent and independent pathways suggest that its entry mechanisms may vary depending on the experimental model, nature of the virions/pseudovirions used and cell type, and in vivo, the level of receptor/serine protease expression might be the determining factor on which pathway is utilized predominantly.

## 3. Role of Other Receptors

Certain cell surface proteins, besides primary receptors, could play a crucial role in aiding coronavirus entry by acting as attachment factors or co-receptors.

### 3.1. Dipeptidyl Peptidase 4 (DPP4)

Also known as CD26, DPP4 serves as the entry receptor for MERS-CoV [13]. It is a type II transmembrane serine exopeptidase expressed in the respiratory tract, liver, kidney, and intestinal cells, and on the surface of immune cells. This protease catalyzes the digestion of multiple chemokines, neuropeptides, and regulatory peptides, and plays crucial metabolic roles, such as regulating glucose levels by inactivating incretins and modulating adenosine levels through its interaction with adenosine deaminase [103,104]. DPP4 is structurally more diverse across species compared to ACE2. Structural studies on DPP4 variants across species demonstrated that MERS-CoV can engage DPP4 from camels, horses, and bats, but not from mice, due to minor structural variations of the DPP4 protein across species [8]. Related bat coronaviruses, such as BatCoV-HKU4, have also shown an affinity for DPP4, suggesting evolutionary adaptations. Although distantly related coronaviruses, like SARS-CoV and SARS-CoV-2, were initially thought not to utilize DPP4 due to differences in the CTD loop structures that affect the receptor-binding interface [105], recent findings, along with the fast pace at which mutations appear in the variants, may suggest otherwise [106].

In silico studies have shown that amino acid residues in the DPP4 can interact with, and are capable of, binding residues in the S RBD, with an affinity of −34.8 kcal/mol, comparable to the binding affinity for ACE2 (−39.2 kcal/mol). The amino acid residues in S RBD that were shown to be involved in the binding with DPP4 were Q498, D405, E484, Y489/N487, N501, and Y505. Mutations in the E484 or adjacent residues were found to be critical for the DPP4-binding ability of SARS-CoV-2 [107]. In regard to variants of SARS-CoV-2, computational analysis revealed that L452R and T478K mutations in the Delta variant directly participate in the interaction with DPP4. Also, the E484K mutation found in Beta and Gamma variants was also found to interact with the receptor [106]. Given that most of those mutations are found in the Omicron variant or its descendants, it is conceivable that further in silico analysis might also reveal similar interactions of the Omicron variants with the DPP4. While in vitro/in vivo evidence of utilization of DPP4 by SARS-CoV-2 remains lacking, clinical studies using DPP4 inhibitors have shown a reduction in respiratory complications, a decrease in mortality, and complications in COVID-19 patients [108,109]. A recent study demonstrated that knockdown of DPP4 in astrocytes and pericytes did not significantly impact infection rates by SARS-CoV-2 (wild-type virus); however, it led to a reduction in the mRNA levels of the N and S proteins compared to the DPP4-expressing control cells, suggesting a decreased replication potential [110].

### 3.2. Neuropilin-1 (NRP1)

Neuropilin-1 (NRP1) is a transmembrane glycoprotein that is involved in many physiological processes, such as axon guidance in the central nervous system, angiogenesis, and immune regulation [111]. While its role in infection by other coronaviruses remains inconclusive, early studies into the pandemic showed that it may act as a direct receptor for SARS-CoV-2 [112] or as an auxiliary factor that enhances viral entry into host cells, particularly in tissues with low ACE2 expression, especially in the CNS [113,114]. An interesting molecular dynamics simulation study [115] showed that the Delta variant exhibited the strongest binding to NRP1, whereas the early Omicron variant BA.1 showed reduced affinity for NRP1 compared to the wild-type. The authors further speculated that viral evolution is unlikely to enhance NRP1 binding affinity, suggesting a potential shift away from the utilization of this receptor.

### 3.3. Aminopeptidase N (APN)

Aminopeptidase N, also termed CD13, is a type II transmembrane metallopeptidase that cleaves amino acids from the N-terminus of peptides. It plays an important role in protein digestion, immune response regulation, angiogenesis, and cellular adhesion. This protein is expressed in epithelial cells of the intestines, the nervous system, and immune cells such as monocytes and dendritic cells. APN is also found in a soluble form in the plasma [116]. It is the primary target receptor for many Alpha coronaviruses, facilitating viral entry through binding to the coronavirus S subunit S1 CTD [117]. Human coronavirus HCoV-229E, transmissible gastroenteritis virus (TGEV), and related feline and canine coronaviruses (FCoV and CCoV) all utilize APN to infect host cells, each adapting distinct molecular interactions to engage the receptors [118]. Interestingly, while APN was initially thought to be unique to Alpha coronaviruses, studies have shown that porcine Delta coronaviruses (PDCoV) can also utilize APN for entry, where the interaction with APN is species-specific, influenced by variations in glycosylation patterns that affect receptor recognition and binding affinity [119]. In regard to Beta coronaviruses and SARS-CoV-2 in particular, there is little evidence to suggest the utilization of this metalloprotease by the virus. Gene expression analysis in human tissue revealed that expression of peptidases such as alanyl and glutamyl aminopeptidases shows a similar pattern to that of ACE2 [120]. Given the similarity in the sequence of the S protein between Alpha and Beta coronaviruses, while it may not be a primary target receptor, for lack of evidence, it is conceivable that it may function as a secondary or auxiliary receptor during infection. Known for its role in the regulation of immune response, others have even hypothesized that APN might influence SARS-CoV-2 pathogenesis through alternative pathways, such as amplifying the immune response, thereby limiting infectivity [121,122].

### 3.4. Glucose-Regulated Protein 78 (GRP78)

As a chaperone protein, besides its main function in protein homeostasis in the endoplasmic reticulum, GRP78 is also involved in many biological processes, such as signaling, inflammation, and cell survival [123]. During the response to ER stress or inflammation, this chaperone protein is also present on the cell surface, where it mediates binding to various ligands [124,125]. At the cell surface, GRP78 was shown to facilitate infection by SARS-CoV-2 through the formation of a complex with the S protein and the host receptor ACE2 [125,126]. In immortalized human monocyte-like THP-1 cells, GRP78 was found to facilitate entry of pseudovirions pseudotyped with the wild-type S protein in an ACE2-independent manner [127]. Utilizing the surface plasmon resonance technique, a research group found direct interaction between GRP78 and the S protein, and the affinity between GRP78 and the S protein of the Omicron variant was approximately two times higher than that of the wild-type. Given that GRP78 is a chaperone protein that is expressed in a wide variety of tissues and cells, this may greatly aid viral entry into non-target/off-target sites.

While there is a lack of studies on this receptor in association with SARS-CoV-2, there is a reason to believe that other variants, such as Delta or the newer emerging Omicron variants, also utilize this molecule for entry, especially given the diversity of COVID-19 symptoms presented by patients infected with those variants.

### 3.5. Carcinoembryonic Antigen-Related Cell Adhesion Molecule 1 (CEACAM1)

CEACAM1, a member of the immunoglobulin superfamily, is expressed mainly in the liver and central nervous system, and is involved in many physiological functions, such as cellular adhesion, development, and immune modulation [128]. The murine coronavirus mouse hepatitis virus (MHV) strain A59, a prototype species for the Betacoronavirus genus, utilizes murine carcinoembryonic antigen-related cell adhesion molecule 1 (mCEACAM1) as its primary receptor [129]. Unlike most coronaviruses that utilize the S protein’s C-terminal domain for receptor binding, MHV-A59 uses the S protein’s N-terminal domain instead (NTD). Structural studies suggest that this NTD usage phenomenon might have evolved from a host-derived galectin domain, retaining a carbohydrate-binding function [130]. To date, there is no evidence to suggest utilization of this protein by SARS-CoV-2 as an entry receptor, although transcriptomic analysis of a limited number of autopsy samples obtained from COVID-19 diseased patients revealed upregulation of CEACAM1, which correlated with viral levels [131]. This certainly does not imply an association between SARS-CoV-2 and CEACAM, and may be a result of immune dysregulation, given that the patients succumbed to COVID-19.

## 4. Other Potential Receptors

Heparan sulfate proteoglycans (HSPGs) are negatively charged macromolecules that are found on the cell surface and extracellular space, facilitating cellular adhesion, division, and migration [132]. In the context of infection with SARS-CoV-2, they were found to act as auxiliary attachment factors that complement ACE2, enhancing the efficiency of SARS-CoV-2’s binding and entry into host cells. This mechanism is particularly important to facilitate viral infection in tissues where expression of ACE2 is low [133,134]. This interaction appears to occur with the S1/S2 interface of each monomer in the trimeric S protein, and at residues 453–459 [135]. Similar interactions have been observed in feline and canine coronaviruses (FCoVs and CCoVs), as well as in human coronaviruses like OC43 and NL63 [7,32]. Other viruses also utilize HSPGs for cell entry, such as HIV, wherein interaction with the HIV envelope glycoprotein gp120 augments viral accumulation at the cell surface, enhancing the binding affinity to CD4 and the co-receptors CCR5 and CXCR4 [136].

Basigin (CD147) is a transmembrane glycoprotein belonging to the immunoglobulin family that is involved in various physiological processes, such as immune response and cell signaling, and has been implicated in assisting entry of pathogens [137]. While direct involvement with SARS-CoV-2 S protein was not proven, it was postulated that it may act as an auxiliary receptor or co-factor for SARS-CoV-2’s S protein, providing an alternative mechanism for viral attachment to the host cell surface, perhaps even independent of ACE2 via activation of the endocytotic pathway [138,139].

C-type lectins, such as DC-SIGN (dendritic cell-specific intercellular adhesion molecule-3) and L-SIGN (liver/lymph node-specific intracellular adhesion molecule), are known to facilitate the entry of HIV-1 by allowing dendritic cells to capture the virus and transfer it to T cells in lymphoid tissues, thereby enhancing infection [140]. In regard to coronaviruses, studies have shown that DC-SIGN and L-SIGN can assist in the entry of various strains, including SARS-CoV, HCoV-229E, and infectious bronchitis virus (IBV), by promoting viral attachment to host cells [141].

Coronaviruses can also exploit toll-like receptors (TLRs) on host cells to enhance infection and trigger immune responses. TLRs, a family of receptors involved in recognizing pathogens, detect viral components like RNA and S proteins, leading to immune activation. Some studies suggest that SARS-CoV-2, along with other coronaviruses, may engage TLRs to support viral entry or amplify inflammatory responses, especially in lung tissues. For example, TLR2 and TLR4 are thought to interact with coronavirus structural proteins, potentially intensifying inflammation, which is common in severe COVID-19 cases. This TLR activation not only aids viral entry, but may also lead to excessive immune responses, contributing to disease severity [142].

More recently, histamine receptor H1 (HRH1) was found to act as an auxiliary receptor for SARS-CoV-2, directly binding to the viral S protein and enhancing ACE2-mediated viral entry [143]. In that study, antihistamines were able to block infection of HEK cells expressing ACE2 by almost all SARS-CoV-2 variants, albeit at higher micromolar concentrations that are not typically achieved with the usual dosage of the drugs [144].

## 5. Receptor-Mediated Endocytotic Pathways

Receptor-mediated endocytosis is a highly specific cellular uptake mechanism, enabling cells to internalize extracellular molecules through the interaction of ligands with their corresponding cell surface receptors. Following binding of the ligands to their receptors, recruitment of intracellular adaptor proteins ensues, triggering a cascade of events that leads to internalization of the particles into cells [145]. The acidic environment of the endosome often facilitates the dissociation of ligands from their receptors, leading to either receptor recycling to the plasma membrane, or trafficking to lysosomes for degradation. While clathrin-mediated endocytosis is the most well-characterized and prevalent pathway, alternative mechanisms also exist, including the caveolae-mediated pathway and clathrin- and caveolin-independent routes [146]. Importantly, several viruses exploit these pathways to gain entry into host cells. Once internalized, these viruses manipulate endosomal maturation and pH-dependent conformational changes to trigger membrane fusion or escape into the cytoplasm, enabling subsequent replication.

Research has shown that caveolae and lipid raft microdomains in the cell membrane play a crucial role in viral endocytosis, facilitating viral entry. Being rich in cholesterol, sphingolipids, and glycosylphosphatidylinositol-anchored proteins, they provide a cholesterol-rich, receptor-dense platform that many viruses utilize for efficient entry and infection. In fact, some viruses, such as certain types of human papillomaviruses [147] and Simian virus 40 (SV40), are known to exploit caveolin-1-dependent pathways for endocytosis [148], while HIV and influenza require the cholesterol-rich domains for membrane fusion and entry [149,150].

Clathrin-mediated endocytosis, on the other hand, is a key cellular process that is also exploited by certain viruses. Under the membrane, clathrin forms a lattice that drives membrane invagination, leading to the formation of clathrin-coated vesicles, thereafter, dynamin mediates vesicle scission, releasing the contents of the vesicles into the cytoplasm followed by fusion with early endosomes [151]. Given that the traffic here is typically directed towards lysosomes, this pathway is favorable for viruses that require low pH for uncoating, such as dengue and adenoviruses [152,153].

In regard to SARS-CoV-2, studies have shown that SARS-CoV-2 utilizes the clathrin-mediated endocytotic pathway after interacting with ACE2 [154,155], although SARS-CoV-2 may also enter cells via clathrin-mediated endocytosis in a manner that is independent of ACE2, potentially involving other receptors.

The transferrin receptor (TFRC or CD71), a membrane protein, plays a key role in binding and internalizing iron-bound transferrin into cells, making it essential for both immune function and cell growth [156]. Recent studies indicate that certain viruses, including the influenza virus, exploit this receptor for cell entry [157,158], although, mechanistic interaction between the virus and the receptor has not yet been explored. More recently, TFRC was identified as a receptor for SARS-CoV-2, facilitating viral entry by directly binding to the S protein in an ACE2 knockdown Calu-3 cell model [159]. To support their findings, the authors utilized soluble anti-TFRC antibodies and specially designed peptides to disrupt the interaction between TFRC and the virus, which was found to abrogate viral infection.

## 6. Mutations in Omicron Subvariants: Implications for Receptor Binding and Viral Entry

It is noteworthy that the Omicron variant contains over 30 mutations, significantly more than any of the previous variants, which has led some to speculate that evolution in a non-human host might be responsible [160]. Mutations such as Q493R, N501Y, and Y505H were found to be particularly well-suited to the ACE2 receptor of mouse origin, rather than that of humans, suggesting a potential mouse origin for the variant.

Novel mutations in the subvariants reflect key changes in the S protein. Mutations such as the “flip mutations”, wherein the amino acid at a given position is “flipped” or changed to a different one, often switching between two possible amino acids, are thought to affect transmissibility, immune escape, or potentially impact vaccine efficacy. Examples of such mutations include L452R, F486V, and Q493R, which are common across several subvariants. These mutations have been shown to significantly impact the variant’s susceptibility to both polyclonal and monoclonal antibodies, enhancing their ability to evade immune responses [161].

Other mutations, such as S371L, and S373P, are also commonly observed in Omicron-related subvariants. Although these mutations were computationally predicted to destabilize the RBD of the S protein in the closed conformation, the cross-RBD interactions that stabilize one RBD in the open conformation within the S trimer, along with compensatory mutations in the ACE2-binding site, may collectively enhance the variant’s affinity to the receptor, likely contributing to increased infectivity [162,163].

The increased transmissibility of newer variants is a complex phenomenon. The enhanced binding affinity of the Omicron RBD to ACE2 may improve the variant’s ability to infect the respiratory tract, where ACE2 expression is relatively low. Mutations such as S477N, N501Y, and T478K have been shown to strengthen the binding to ACE2 [164], which may, in turn, shift the viral tropism, promoting infection and replication in the upper respiratory tract and accelerating the spread of the infection [165]. In addition, mutations that facilitate immune evasion contribute significantly to the rapid spread and dominance of Omicron and subsequent variants over earlier strains. Studies have demonstrated that Omicron exhibits substantial resistance to neutralizing antibodies, with both post-convalescent and post-vaccination sera showing significantly reduced neutralization efficacy compared to previous variants [166].

Further evidence suggests that Omicron has altered its preferred cell entry mechanism, shifting from TMPRSS2-mediated surface fusion of the spike protein to cathepsin-mediated endosomal fusion, independent of TMPRSS2 [167]. This adaptation may indicate a shift in pathogenicity, potentially mitigating severe lung infection. In a study assessing the infection of mice and hamsters with several Omicron subvariants, the authors found reduced infection in immunocompetent and human ACE2-expressing mice and hamsters compared to previous SARS-CoV-2 variants [168]. Despite evidence that the Omicron S protein binds more strongly to mouse ACE2, the infected transgenic mice and hamsters showed limited pathogenesis and lower viral loads in both upper and lower respiratory tracts, hinting at the course of mild infections. This observation is consistent with current clinical data, which indicates that the majority of current COVID-19 cases present with mild symptoms and reduced severity compared to earlier variants. Overall, the immune evasion mechanisms and the shift in Omicron’s cell entry pathways likely contribute to its reduced pathogenicity, as it induces less severe disease while maintaining high transmissibility.

A summary of the most recent SARS-CoV-2 variants, along with their significant mutations in the S protein, is provided in Table 3. These mutations highlight the ongoing evolution of the virus and underscore their potential impact on transmissibility, and immune escape.

## 7. Evidence of Shift in Receptor Utilization from the Changing Clinical Spectrum of Infection

To date, it is evident that the pattern of symptoms associated with COVID-19 varied over time as the pandemic progressed, with shifts in the symptom clusters linked to different viral variants (Table 4). During the initial phase, infection with the prototypic Wuhan strain typically involved the lower respiratory tract, resulting in severe pneumonia, dyspnea, and acute respiratory distress syndrome (ARDS) [170]. The hallmark of early COVID-19 during the pandemic was the reduced or lost sense of smell and taste. While nearly two-thirds of patients infected with the Alpha and Delta variants reported these symptoms, their prevalence significantly decreased with Omicron. Evidence suggests that the lack of anosmia or ageusia is associated with the Omicron variant’s decreased affinity for the olfactory epithelium [170]. As the pandemic advanced, common upper respiratory cold-like symptoms such as nasal congestion, sneezing and sore throat became more prevalent with the Delta, Omicron, and its subsequent subvariants [170,173], in fact, studies demonstrated higher viral loads in the upper respiratory tract, particularly in the nasal cavity, trachea, and throat, compared to the lower respiratory tract [174]. Moreover, long-term sequelae of COVID-19 were more frequently reported after infection with the prototypical and earlier Alpha and Delta variants, more so compared to the more recent Omicron and its subvariants [175]. This difference is likely attributable to the heightened inflammatory response and greater severity of infection associated with the earlier variants, rather than differences in tissue tropism.

## 8. Therapeutic Implications

As previously discussed, mutations in the S protein have greatly impacted viral infectivity and immune evasion, most importantly, altering the effectiveness of therapeutic interventions, particularly susceptibility to monoclonal antibodies (mAbs). This necessitated the development of updated vaccine formulations to target both ancestral and emerging variants, resulting in a constant “catch-up” scenario in vaccine development, with the virus maintaining an advantage. The updated 2024 to 2025 formula mRNA COVID-19 vaccines are still based on the S protein of KP.2, which has already become outdated [179].

The emerging Omicron subvariants, harboring variant-specific critical mutations in the RBD such as S371L, D405N, K417N, N440K, G446S, F486V, E484K/A, and Q493R, were shown to reduce the neutralization potency of several mAbs by altering key antigenic sites in vitro, which may translate into therapeutic escape [180]. Tixagevimab/Cilgavimab (Evusheld) is a monoclonal antibody cocktail that gained emergency authorization in December of 2021 for the treatment and pre-exposure prophylaxis of COVID-19 in individuals who are immunocompromised or otherwise unable to mount an adequate immune response to vaccination. In the face of emerging Omicron variants, clinical and in vitro data showed a significant loss of efficacy, especially against BA.4/BA.5, BA.2, BQ.1, XBB.1.5, XBB.1.16, and EG.5 variants [181,182], which led to the loss of FDA authorization for use in the United States [183].

Pemivibart (Pemgarda) is a more recent monoclonal antibody that was granted Emergency Use Authorization (EUA) by the FDA in August of 2024, and so far had demonstrated notable efficacy in neutralizing BA.1, BA.2, BA.5, BQ.1.1, XBB.1.5, and JN.1-lineage variants (KP.2/KP.3/KP.3.1.1/XEC/LP8.1), with EC_50_ values ranging from 0.198 to 14.3 nM [184].

Research on hybrid immunity (a combination of infection and vaccination) has provided compelling insights. The SC27 monoclonal antibody, presumed to be generated through mRNA vaccination in an infection-naïve individual, showed a marked enhancement in potency following a hybrid immune response triggered by a subsequent infection. This antibody was found to effectively neutralize both the prototypical and current variants of concern of SARS-CoV-2, as well as a range of antigenically distinct zoonotic sarbecoviruses that may pose a future risk to humans. Its neutralization potency, as measured by IC_50_ values, ranged from 13 to 264 ng/mL [185]. It remains in the research and development phase, and as of the time of writing this review, it has not yet received EUA or full approval for widespread clinical application.

Given the pivotal role of the ACE2 receptor in mediating infectivity by SARS-CoV-2, the idea of using decoy ACE2 (soluble serum form) to target the virus’s S protein, preventing it from attaching to human cells gained traction earlier on at the start of the pandemic [186,187]. Utilizing minimal receptor fragments that mimic the host ACE2 receptor without triggering an immune response, however, is key. The challenge lies in identifying the most efficient minimal sequences of ACE2 that retain the ability to bind the virus while avoiding immune recognition. Mutations in the RBD of current variants, or perhaps the shift toward a completely different primary receptor, have cast doubt on that strategy. However, through computational design, an affinity-enhanced ACE2 decoy demonstrated strong binding to both the SARS-CoV-2 Delta and Omicron variants, suggesting its potential as an effective therapeutic against a wide range of SARS-CoV-2 variants and other sarbecoviruses [188], including BQ.1 and XBB.1 [189]. Beyond ACE2 affinity modulation, the evidence pointing to the utilization of alternative receptors, such as neuropilin-1 and heparan sulfate, to facilitate infection by SARS-CoV-2 highlights the urgent need for surveillance of S protein evolution and the development of broadly neutralizing antibodies targeting conserved viral epitopes to mitigate the risk of immune escape and receptor switching.

## 9. Conclusions

The wide variability in receptor utilization by coronaviruses underscores their adaptability and capacity to cross species barriers. In this review, we explored the receptors utilized by the SARS-CoV-2 spike protein and their impact on infectivity and tissue tropism across variants, which collectively shaped the trajectory of the COVID-19 pandemic. This shift in receptor utilization could explain the wide clinical spectrum of infection by different variants throughout the pandemic. Studies on the effects of mutations on the interaction between the S protein and potential receptors are of great importance to understanding the entry pathways of coronaviruses. A deeper understanding of how emerging variants adapt to different receptors is crucial for predicting tissue tropism, clinical outcomes, and transmission patterns. Furthermore, such knowledge will undoubtedly lay a critical foundation for developing novel therapeutic strategies and preventive interventions targeting viral entry mechanisms. There are indeed broader implications from a public health perspective, vaccine development, and pandemic preparedness. Understanding viral evolution and receptor interactions would undoubtedly aid in the robust and adaptable vaccine designs, offering broader protection against emerging strains. Moreover, this knowledge could improve surveillance efforts, allowing for more rapid identification of emerging variants. The shifting of receptor utilization dynamics highlights the need for adaptive strategies. Ongoing and future research could benefit greatly from using structural and molecular modeling tools to predict the interaction between the S protein and current/potential receptors across coronaviruses and other viral families. Such tools would enable identification and analysis of viral variants, receptor mapping and interaction modeling. In this era of artificial intelligence, AlphaFold and machine learning algorithms, would further enhance predictive modeling, and help pinpoint critical sites that would enhance the development of vaccines targeting viral entry.

## Figures and Tables

**Figure 1 viruses-17-00691-f001:**
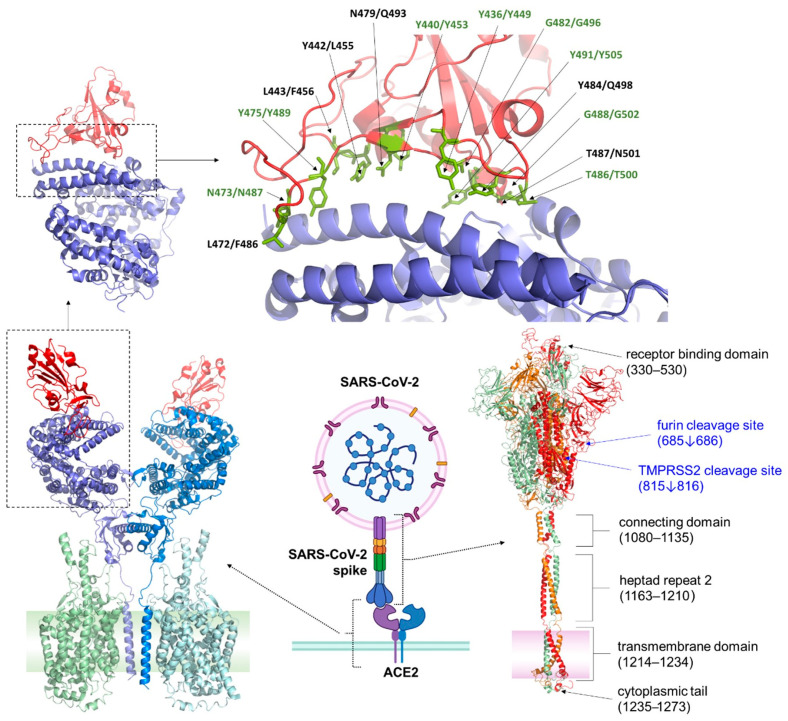
Structures of SARS-CoV-2 S protein and ACE2 receptor. Lower middle panel: Interaction of SARS-CoV-2 S protein with ACE2 receptor. The figure was prepared using BioRender (https://www.biorender.com/, accessed on 16 March 2025). Lower left panel: Structure of ACE2 is shown based on cryo-electron microscopy (PDB ID: 6M17) [37]. The monomers of the ACE2 homodimer (blue color) are complexed with sodium-dependent neutral amino acid transporter B^0^AT1 (green). The schematic representation of the cellular membrane is also shown in green. The RDB of SARS-CoV-2 S protein binding to the ACE2 is shown in red. Lower right panel: The structure of the full-length SARS-CoV-2 S protein based on a modeled structure [60]. The subunits of the homotrimeric complex have green, orange, and red colors. The schematic representation of the virion’s membrane is also shown in pink color. Some domains of the S protein are labeled [60], and furin and TMPRSS2 cleavage sites are also shown. Upper panels: The complex of ACE2 and SARS-CoV spike’s RBD is shown based on a crystal structure (PDB ID: 2AJF) [11]. The ACE2 has blue, while the RBD has red color. An enlarged view of the ACE2-RBD complex (shown in the upper left panel) is represented in the upper right panel. The residues contacting ACE2 and RBD are shown by green sticks. The residue numbers are shown for SARS-CoV and SARS-CoV-2 RBDs, as well, the residue names being identical in these viruses are labeled in green, based on literature data [54].

**Figure 2 viruses-17-00691-f002:**
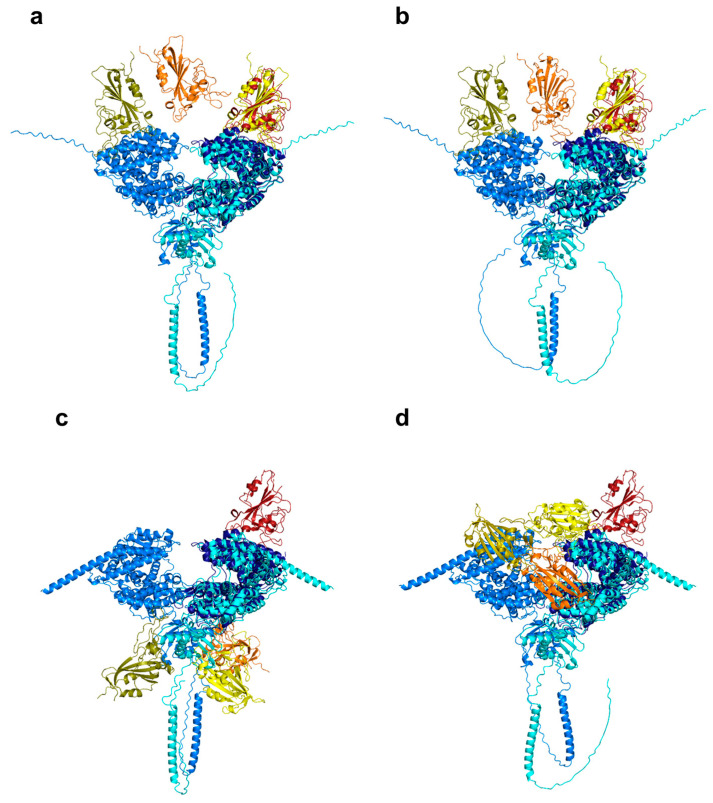
Structures of the predicted ACE2 dimer with RBD domains of the SARS-CoV-2 S protein trimer, compared to the aligned experimentally determined ACE2-RBD complex (PDB ID: 2AJF) [54]. (**a**) Wild-type RBD, correctly bound conformer. (**b**) XEC variant RBD, correctly bound conformer. (**c**) XEC variant RBD, incorrectly bound conformer. (**d**) Wild-type RBD, top conformer, derived with the GRAMM webserver. The protomers of the predicted ACE2 dimer are colored with light blue and cyan, while the experimentally derived ACE2 is colored in navy blue. The three predicted RBD subunits are colored olive, orange, and yellow, respectively; the experimentally determined RBD subunit is colored red.

**Table 1 viruses-17-00691-t001:** Human pathogenic coronaviruses and their primary receptors.

Virus	Genus	Primary Receptors	Year of Outbreak	Disease Caused
HCoV-229E	Alphacoronavirus	APN (Aminopeptidase N) [29]	Identified in 1960s	Common cold, respiratory infections
HCoV-OC43	Betacoronavirus	9-O-acetylsialic acids as receptor [26]	Identified in 1960s	Common cold, bronchitis, pneumonia
SARS-CoV-1	Betacoronavirus	ACE2 [30]	2002–2003	Severe acute respiratory syndrome (SARS)
HCoV-NL63	Alphacoronavirus	ACE2 [27]	2004	Common cold, croup, bronchiolitis
HCoV-HKU1	Betacoronavirus	9-O-acetylsialic acids as receptor [26]	2005	Common cold, pneumonia
MERS-CoV	Betacoronavirus	DPP4 (CD26) [31]	2012	Middle East respiratory syndrome (MERS)
SARS-CoV-2	Betacoronavirus	ACE2 [32]	2019–2020	Coronavirus disease of 2019 (COVID-19)

**Table 2 viruses-17-00691-t002:** Key mutations in the RBD of the S protein in SARS-CoV-2 variants [66].

Pango Lineage	Variant	Key Mutations in RBD
B.1.1.7	Alpha	N501Y
B.1.351	Beta	K417N, E484K, N501Y
P.1	Gamma	K417T, E484K, N501Y
B.1.617.2	Delta	L452R, T478K
B.1.427/B.1.429	Epsilon	L452R
P.2	Zeta	E484K
B.1.525	Eta	E484K
P.3	Theta	E484K, N501Y
B.1.526	Iota	E484K
B.1.617.1	Kappa	L452R, E484Q
C.37	Lambda	L452Q, F490S
B.1.621	Mu	R346K, E484K, N501Y

**Table 3 viruses-17-00691-t003:** Key Omicron subvariants and their characteristic mutations in the S protein. Data was collected from [66,169,170,171,172]. * Cumulative prevalence is the ratio of the sequences collected since the identification of the variant based on data from GISAID Initiative.

Variant	Lineage	Key Spike Mutations in the RBD	Cumulative Prevalence Worldwide *
KP.2	Descendant of BA.4 lineage of the Omicron variant	I332V, G339H, R346T/K, K356T, S371F, S373P, S375F, T376A, R403K, D405N, R408S, K417N, N440K, V445H, G446S, N450D, L452W/R, L455S, F456L, N460K, S477N, T478K, N481K, del483/483, E484K, F486P/V, Q493R, Q498R, N501Y, Y505H	2% as of October 2024
KP.3	Evolved from the BA.2.12.1 subvariant	I332V, R346K, K356T, S371F/L, S373P, S375F, T376A, G339H, R403K, D405N, R408S, K417N, N440K, V445H, G446S, N450D, L452W/R, E554K, L455S, F456L, N460K, S477N, T478K, N481K, V483del, E484K, F486P/V, Q493E/R, Q498R, N501Y, Y505H	27% as of December 2024
LB.1	Descendant of the BA.1 Omicron subvariant	I332V, G339H, R346T, K356T, S371F, S373P, S375F, T376A, R403K, D405N, R408S, K417N, N440K, V445H, G446S, N450D, L452W, L455S, F456L, N460K, S477N, T478K, N481K, del483/483, E484K, F486P, Q498R, N501Y, Y505H	3% as of December 2024
BA.2.86	Descendant of BA.2 Omicron variant	G339H, K356T, S371F, S373P, S375F, T376A, R403K, D405N, R408S, K417N, N440K, V445, G446S, N450D, L452W, N460K, S477N, T478K, N481K, V483Δ, E484K, F486P, Q498R, N501Y, Y505H	27% worldwide as of October 2024
XBB.1.5	Recombination between BA.2.10.1 and BA.2.75 subvariants	G339H, R346T, L368I, S371F/L, S373P, S375F, S477N, T376A, D405N, R408S, S413R, K417N, N440K, V445P, G446S, L452R, N460K, T478R, E484A, F486P, F490S, S494P, Q498R, N501Y, Y505H	8% as of October 2024
JN.1	Subvariant of Omicron variant BA.2.86	G339D, K356T, S371F, S373P, S375F, T376A, R403K, D405N, R408S, K417N, N440K, G446S, L452W/R, N450D, L455S, N460K, F486S, V445H, Y505H	27% as of December 2024
XEC	Recombinant variant combining genetic material from the KS.1.1 and KP.3.3 subvariants	I332V, G339H, R346K, K356T, S371F/L, S373P, S375F, T376A, R403K, D405N, R408S, K417N, N440K, V445H, G446S, N450D, L452W/R, L455S, F456L, N460K, S477N, T478K, N481K, del483/483, E484K, F486P/V, Q493E/R, Q498R, N501Y, Y505H	1% as of October 2024

**Table 4 viruses-17-00691-t004:** Common and prominent symptoms of SARS-CoV-2 variants [176,177,178].

	Common Symptoms	Prominent Symptoms
**Original Strain (Wuhan)**	Fever, cough, fatigue, shortness of breath, muscle aches	Anosmia, ageusia, pneumonia, ARDS
**Alpha (B.1.1.7)**	Fever, headache, vertigo, nasal congestion, throat pain, dyspnea, nausea, diarrhea	Anosmia, ageusia, delerium, depression, pneumonia, ARDS
**Beta (B.1.351)**	Fever, headache, vertigo, nasal congestion, throat pain, dyspnea, nausea, diarrhea	Anosmia, ageusia
**Delta (B.1.617.2)**	Fever, fatigue, muscle ache, diaphoresis, nasal congestion, dyspnea, diarrhea	Headache, throat pain, vomiting, diarrhea, chest pain, pneumonia, ARDS
**Omicron (B.1.1.529)**	Fever, cough, fatigue, nasal congestion, muscle aches,	Throat pain, vomiting, diarrhea, vertigo
**Omicron Subvariants (BA.4/BA.5/XEC/JN.1)**	Rhinorrhea, cough, fatigue, headache, fever, muscle aches	Sore throat, nasal congestion

## Data Availability

Not applicable.

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
