# Peer review of "Receptor Binding for the Entry Mechanisms of SARS-CoV-2: Insights from the Original Strain and Emerging Variants"

_viruses, 2025, doi:10.3390/v17050691_

Round 1
Reviewer 1 Report
Comments and Suggestions for Authors
This review provides a comprehensive overview of SARS-CoV-2 receptor binding and entry mechanisms, focusing on ACE2 as the primary receptor and exploring alternative pathways (e.g., TMPRSS2-dependent vs. cathepsin-mediated entry). The authors analyze how spike protein mutations in emerging variants (e.g., Delta, Omicron subvariants) alter receptor affinity, tissue tropism, and clinical outcomes. Structural modeling and in silico predictions are used to compare RBD-ACE2 interactions across variants. The review also discusses therapeutic implications, including monoclonal antibodies and decoy receptor strategies. Clinical symptom shifts across variants are linked to evolving receptor utilization.
This is a well-structured review that combines complex virological and clinical data. However, the paper would be more appropriate for publication by addressing the following points.
First, the function of spike D614G mutation is not well described. This mutation is far from furin cleavage site, however, the authors claimed that “ it indirectly enhances the proteolytic processing by furin at the S1/S2 site by furin” on line 256. According to the protein structure in PMID: 32991842, D614G mutation keeps RBD in the open conformation, which sounds more reasonable.
Secondly, the discussion of DPP4 and GRP78 as potential receptors (Sections 3.1 and 3.4) relies heavily on computational predictions. Cite experimental validations where available.
Last, there are some typos or incorrect information. It claims E484K mutation found in Alpha and Gamma variants on line 101, where it should be Beta rather than Alpha variant. Typographical error: In the "List of abbreviations", "FCoVs: Fline coronaviruses" should be corrected to "FCoVs: Feline coronaviruses".
Author Response
We thank the reviewer for their valuable time and constructive feedback on our manuscript. We will hereby try to address your comments to the best of our ability.
1) First, the function of spike D614G mutation is not well described. This mutation is far from furin cleavage site, however, the authors claimed that “ it indirectly enhances the proteolytic processing by furin at the S1/S2 site by furin” on line 256. According to the protein structure in PMID: 32991842, D614G mutation keeps RBD in the open conformation, which sounds more reasonable.
- The paragraph regarding the D614G mutation was complemented to further elaborate on the effects of this mutation:
“D614G that is observed in Alpha, Beta, Gamma, Delta and all current Omicron variants enhances S stability, and indirectly enhances the proteolytic processing by furin at the S1/S2 site, further facilitating viral adaptation and natural selection [86,87]. In single-cycle infection assays [88], the D614G mutation was shown to increase the stability of the spike protein's S1/S2 interface. This enhanced stability, in turn, created a permissive environment for the acquisition of additional mutations that promoted more efficient cleavage at the S1/S2 site, ultimately driving the evolution of SARS-CoV-2 variants with highly fusogenic S proteins. The authors were able to corroborate previous findings indicating that increased S protein stability facilitates viral entry through the cathepsin L-dependent pathway [89].”
2) Secondly, the discussion of DPP4 and GRP78 as potential receptors (Sections 3.1 and 3.4) relies heavily on computational predictions. Cite experimental validations where available.
- In vitro studies on the role of DPP4 in SARS-CoV-2 infection are indeed lacking, however, we have found a recent study on SARS-CoV-2 infection of the astrocytes and pericytes. The paragraph is now complemented with the following:
“A recent study demonstrated that knockdown of DPP4 in astrocytes and pericytes did not significantly impact infection rates by SARS-CoV-2 (wild-type virus); however, it led to a reduction in the mRNA levels of the N and S proteins compared to the DPP4-expressing control cells, suggesting a decreased replication potential [110].”
In regards to the GRP78, we found a study performed in THP-1 cells, and this has now been included in the subsection:
“In immortalized human monocyte-like THP-1 cells, GRP78 was found to facilitate entry of pseudovirions pseudotyped with the wild-type S protein in an ACE2-independent manner [127]”
Unfortunately, studies on the role of this receptor in the context of SARS-CoV-2 infection are still lacking.
3) Last, there are some typos or incorrect information. It claims E484K mutation found in Alpha and Gamma variants on line 101, where it should be Beta rather than Alpha variant.
- Thank you for pointing this out. We apologize for the mistake and it has now been corrected. While this mutation appeared in some later sub-lineages, it was not present in the Alpha variant.
4) Typographical error: In the "List of abbreviations", "FCoVs: Fline coronaviruses" should be corrected to "FCoVs: Feline coronaviruses".
- Thank you. The mistakes have now been ironed out.
Reviewer 2 Report
Comments and Suggestions for Authors
Mohamed Mahdi and colleagues reviewed the receptor binding evolution of SARS-CoV2 making several implications on pathogenic features, viral therapy and vaccines. The matter of the paper is of foremost interest and it is well done. I have only few points to improve.
1- I suggest to add a table with the RBD mutation evolution from original Wuhan to omicron variant (including alpha, beta, delta and other). This could be help the reader to understand the receptor binding mutation evolution in the spike.
2- I suggest to change the title with “Receptor Binding for the Entry Mechanisms of SARS-CoV-2: 2 Insights from the Original Strain and Emerging Variants”. This in line of the text that approached the receptor binding and not the entry mechanism (fusion and other steps).
3- Line 144: “The spike (S)...” it has been previously used. I suggest to use only S or only spike.
4- In table 1, family should be corrected with genus.
5- All typos should be corrected.
Author Response
We thank the reviewer for their careful evaluation and insightful comments which have helped improve the quality of our manuscript.
1- I suggest to add a table with the RBD mutation evolution from original Wuhan to omicron variant (including alpha, beta, delta and other). This could be help the reader to understand the receptor binding mutation evolution in the spike.
- Thank you for this valuable suggestion. We have included Table 2, which summarizes key mutations in the RBD of SARS-CoV-2 variants.
2- I suggest to change the title with “Receptor Binding for the Entry Mechanisms of SARS-CoV-2: 2 Insights from the Original Strain and Emerging Variants”. This in line of the text that approached the receptor binding and not the entry mechanism (fusion and other steps).
- We agree with the reviewer. Thank you for the suggestion. The title has now been modified.
3- Line 144: “The spike (S)...” it has been previously used. I suggest to use only S or only spike.
- We apologize for the mistake. “spike” has now been replaced with “S”
4- In table 1, family should be corrected with genus.
- This has now been corrected. Thank you.
5- All typos should be corrected.
- All the typos and grammatical errors have been ironed out. Thank you.
Reviewer 3 Report
Comments and Suggestions for Authors
This work of Mahdi et al. is a nice review on receptor binding and entry mechanisms of SARS-CoV-2 ranging from wild-type to various new variants. Also the connection to other coronavirus is relevant and of interest. For the topic it may be worthwhile to discuss a bit in more detail the conflicting phenomena of increased infectiousness of Omicron variants but decreases pathogenicity compared to wild-type variant or earlier VoCs (correlating with reduced cell fusion).
There is a major point of scepticism which should be addressed: this regards the affected (reduced) binding of JN.1 and XEC variants RBD to ACE2 dimer as predicted by using AlphaFold3. This is a somewhat contra-intuitive result since the Omicron variants are known to have higher binding affinity to ACE2 receptor compared to wild-type. The reviewer recommends to confirm the results of the AI (Alphafold3) data by other docking models, such as e.g. HDOCK and GRAMM.
Author Response
1) For the topic it may be worthwhile to discuss a bit in more detail the conflicting phenomena of increased infectiousness of Omicron variants but decreases pathogenicity compared to wild-type variant or earlier VoCs (correlating with reduced cell fusion).
We thank the reviewer for their insightful comments and for taking the time to review our manuscript.
- The paragraph has now been expanded to highlight the increased transmissibility and decreased pathogenesis of the omicron variant:
“The increased transmissibility of newer variants is a complex phenomenon. The enhanced binding affinity of the Omicron RBD to ACE2 may improve the variant's ability to infect the respiratory tract, where ACE2 expression is relatively low. Mutations such as S477N, N501Y, and T478K have been shown to strengthen the binding to ACE2 [164], which may, in turn, shift the viral tropism, promoting infection and replication in the upper respiratory tract and accelerating the spread of the infection [165]. In addition, mutations that facilitate immune evasion contribute significantly to the rapid spread and dominance of Omicron and subsequent variants over earlier strains. Studies have demonstrated that Omicron exhibits substantial resistance to neutralizing antibodies, with both post-convalescent and post-vaccination sera showing significantly reduced neutralization efficacy compared to previous variants [166]. Further evidence suggests that Omicron has altered its preferred cell entry mechanism, shifting from TMPRSS2-mediated surface fusion of the spike protein to cathepsin-mediated endosomal fusion, independent of TMPRSS2 [167]. This adaptation may indicate a shift in pathogenicity, potentially mitigating severe lung infection. In a study assessing the infection of mice and hamsters with several Omicron subvariants, the authors found reduced infection in immunocompetent and human ACE2-expressing mice and hamsters compared to previous SARS-CoV-2 variants [168]. Despite evidence that the Omicron S protein binds more strongly to mouse ACE2, the infected transgenic mice and hamsters showed limited pathogenesis and lower viral loads in both upper and lower respiratory tracts, hinting at the course of mild infections. This observation is consistent with current clinical data, which indicate that the majority of current COVID-19 cases present with mild symptoms and reduced severity compared to earlier variants. Overall, the immune evasion mechanisms and the shift in Omicron's cell entry pathways likely contribute to its reduced pathogenicity, as it induces less severe disease while maintaining high transmissibility.”
2) There is a major point of scepticism which should be addressed: this regards the affected (reduced) binding of JN.1 and XEC variants RBD to ACE2 dimer as predicted by using AlphaFold3. This is a somewhat contra-intuitive result since the Omicron variants are known to have higher binding affinity to ACE2 receptor compared to wild-type. The reviewer recommends to confirm the results of the AI (Alphafold3) data by other docking models, such as e.g. HDOCK and GRAMM.
Thank you for the suggestion. On the reviewer’s request, we have applied the GRAMM webserver to dock the RBD trimer to the ACE2 dimer. Out of the predicted structures, only in case of the wild type RBD did we observe a conformation that approximates the experimentally derived bound RBD conformer. The method and the results were added to the main text.
Round 2
Reviewer 3 Report
Comments and Suggestions for Authors
The revised manuscript can be recommended for publication